# Radiofrequency Ablation for Locoregional Structural Incomplete Response in Differentiated Thyroid Cancer: Initial Experience in Greece

**DOI:** 10.3390/biomedicines13020255

**Published:** 2025-01-21

**Authors:** George Simeakis, Aikaterini Kapama, Rodis D. Paparodis, Pyrros Gkousis, Panayiotis Koursaros, Christos Kokkinis, Maria Zozolou, Myrsini Gkeli

**Affiliations:** 1Endocrine Department—Thyroid Cancer Outpatient Clinic, 401 General Military Hospital of Athens, 11525 Athens, Greece; a.kapama@gmail.com; 2Division of Endocrinology, Diabetes and Metabolism, Loyola University Medical Center, Edward Hines Jr. VA Hospital, Hines, IL 60153, USA; rodis@paparodis.gr; 3Department of Radiology, Saint Savvas, Anticancer Oncology Hospital of Athens, 11522 Athens, Greece; pyrrosgkousis@hotmail.com (P.G.); panagiotis-koursaros@hotmail.com (P.K.); chriskok1995@gmail.com (C.K.); myrgel@gmail.com (M.G.); 4Athens Vision Eye Institute, 17673 Athens, Greece; m.zozolou@gmail.com

**Keywords:** differentiated thyroid cancer, locoregional recurrence, minimally invasive treatments, radiofrequency ablation, radioiodine refractory

## Abstract

**Background/Objectives:** Structural incomplete response (SIR) (persistence/recurrence) may occur in 2–6% of low-risk differentiated thyroid cancer (DTC)-cases and in 67–75% of high risk. Regarding locoregional disease, surgery is the optimal therapeutic modality if the smallest dimension of the targeted node is ≥8 mm or ≥10 mm (central or lateral compartment). In the presence of smaller nodes, contraindications or the patient’s unwillingness for reoperation, active surveillance (AS) or minimally invasive treatments (MITs) may be considered. **Methods:** We retrospectively studied eight DTC patients with SIR confirmed by ultrasound (U/S)-guided fine-needle aspiration cytology (FNAC) and the measurement of Thyroglobulin (Tg) in the washout fluid. Fourteen malignant lesions were ablated by radiofrequency (RF). We assessed prior to RF ablation (RFA) and consecutively at one month, three months and, then, every three months the volume of each lesion, serum Tg and Anti-Tg antibodies and calculated the volume reduction ratio (VRR). **Results:** Patients were followed for a mean period of 13.25 months (range: 4–24) after RFA was performed. The targeted lesions reduced significantly from a median volume of 0.24 mL (range: 0.09–0.9) to 0.02 mL (range: 0–0.03) (*p* < 0.05), with a median VRR of 94.5% (range: 78–100%) and concomitant significant biochemical remission (decrease in serum Tg from a median of 1.05 ng/mL to 0.2 ng/mL, *p* < 0.05). In one patient with an aggressive radioiodine (RAI)-refractory histological variant, re-recurrence was documented, which was successfully re-ablated by RF. In two patients, Horner syndrome was diagnosed as an RFA complication, which was totally resolved within six months. **Conclusions**: RFA may be considered as an effective and safe MIT in selective DTC patients with SIR, especially in cases of smaller lesions. Additional prospective studies are needed, including aggressive DTC histological variants towards a tailored therapeutic approach.

## 1. Introduction

Thyroid cancer (TC) is the most common endocrine malignancy although it accounts for only 2.2% of all new cancer cases in the United States [1]. The incidence rate of TC has been increasing until recently, a fact that could be partially attributed to the extended use of diagnostic imaging tools such as ultrasound (U/S), computed tomography (CT) or magnetic resonance imaging (MRI), where even small thyroid nodules are incidentally detected during the medical work-up of a disease other than thyroid cancer [2,3,4]. However, since 2014, a decreasing trend has been observed, which is due to the adoption of stricter diagnostic criteria. Nevertheless, the number of new cases and deaths remains relatively high. It is estimated that, in 2024, new thyroid cancer cases in the United States will be up to 44,020 (12,500 in men and 31,520 in women), while deaths will be up to 2170 (990 in men and 1180 in women) [5,6].

The majority of TC cases (~90%) arise from thyroid follicular cells with the well-differentiated thyroid carcinoma (WDTC) being the most common pathological entity. WDTC is further divided, based on histopathological criteria, into two main subtypes, papillary thyroid carcinoma (PTC, 75–80% of cases) and follicular (FTC, 8–10% of cases) [7,8]. In the updated World Health Organization (WHO) 2022 classification of thyroid neoplasms, the invasive encapsulated follicular variant of papillary thyroid carcinoma (IEFV—PTC) and oncocytic carcinoma (OCA), which, based on the previous WHO 2017 classification, was called Hürthle cell, are described as autonomous pathological entities [9,10]. Prognosis of WDTC is favorable with the 5-year relative survival rate (data from 2012–2018) being up to 98% even in cases of locoregional metastatic disease, while in cases of distant metastases, this falls to 67–74% [11,12]. FTC tends to behave more aggressively than PTC, presenting more frequently vascular invasion and distant metastases, thus characterized by poorer prognosis [13,14,15]. Other types of thyroid carcinomas include medullary thyroid carcinoma (MTC) (~3–5% of thyroid carcinomas), which arises from the parafollicular C-cells of the thyroid gland, differentiated high-grade thyroid carcinoma (DHGTC), which compromises a new pathological entity with intermediate prognosis, poorly differentiated (PDTC) (2–5%), and anaplastic thyroid carcinoma (ATC) (~1%), both characterized by a relatively poor prognosis [16,17,18].

Albeit WDTC is characterized by an excellent prognosis, structural incomplete response (SIR) to the initial treatment (i.e., surgery and the selective use of radioiodine (RAI) ablation) may be present in 2–6% of American Thyroid Association (ATA) low-risk patients, 19–28% of ATA intermediate risk, and up to 75% of ATA high risk, thus leading to additional treatments and active monitoring [19,20,21,22]. The term SIR may refer to disease persistence or recurrence, and, thus, appropriate discrimination should be applied regarding these two different entities. According to the largest study to date, which managed to elucidate the differences between persistence and recurrence in TC, persistence was defined as the “presence of disease ab initio since diagnosis” while true recurrence was defined as a “relapse after being 12 months disease-free”, with the two conditions being characterized by different clinical outcomes [23]. Local persistence or recurrence is present in 5–7% of WDTC patients with surgical central or/and lateral neck dissection constituting the optimal therapeutic approach, under the condition of biopsy-proven disease [19,24,25,26,27]. Surgery is particularly recommended for central neck nodes ≥ 8 mm and lateral neck nodes ≥ 10 mm in the smallest diameter that can be localized on anatomic imaging [19]. On the other hand, in cases of smaller or/and iodine refractory local metastases, patient reluctance or inability to undergo revision surgery, active surveillance (AS) and non-surgical, minimally invasive treatments (MITs) should be considered [19,28]. These may include, but are not limited to, ethanol ablation, laser ablation, radiofrequency ablation (RFA), microwave ablation, cryoablation and high-intensity-focused U/S [19,24].

RFA is a thermal ablation technique that can be used to destroy neoplastic tissue by increasing the intra-tumoral temperature to more than 55 °C, through an electrode tip that combines frictional and conduction heat, generated from high-frequency alternating electric current oscillating between 200 and 1200 kHz [25,29]. It was introduced as an alternative therapeutic modality to treat the local recurrence of WDTC by Dupuy in 2001 with very promising results [30]. Currently, RFA has gained increasing interest for the treatment both of benign thyroid nodules (including autonomously functioning thyroid adenomas) to overcome symptomatic disease or cosmetic issues as well as of malignant thyroid tissue (micro-PTC or local persistence/recurrence) in selective cases [31,32,33,34]. The use of RFA as palliative treatment is also under consideration in a few cases of advanced MTC and ATC [26,29,35]. Regarding cervical WDTC recurrence, RFA has emerged as a reasonable therapeutic modality showing adequate efficacy in numerous studies, albeit the majority of them included small patient cohorts [32,33,34,36,37,38,39]. It is characterized by a low complication rate of 2.38%, with minor adverse events such as pain, hematoma, vomiting, skin burns, and transient thyroiditis. Major complications are rare and may include dysphonia (permanent or transient), nodule rupture, permanent hypothyroidism, and brachial plexus injury [40]. Other serious complications may result from injury to the esophagus, trachea, and other nerves such as sympathetic ganglion, spinal accessory and phrenic nerves [41,42]. As data regarding RFA’s efficacy in recurrent thyroid cancer are still limited coming mainly from single-center studies, it has not yet been established as a widely recommended therapeutic modality by clinicians, especially in Western Europe and the United States [26,34]. The aim of this retrospective study is to present eight DTC cases with locoregional SIR (persistent or recurrent disease) and evaluate for the first time in Greece the safety and efficacy of RFA as an alternative treatment modality for locoregional disease control.

## 2. Materials and Methods

### 2.1. Subjects

A total number of 8 patients (6 women) were retrospectively studied between June 2021 and December 2024 after they were referred to the Thyroid Cancer Outpatient Clinic of the 401 General Military Hospital of Athens due to locoregional SIR as detected by high-resolution U/S and confirmed by U/S-guided fine-needle aspiration cytology (FNAC) and the measurement of Thyroglobulin (Tg) in the washout fluid. Main inclusion criteria were as follows: personal history of DTC as confirmed by histopathological examination with disease persistence or recurrence following initial treatment (i.e., total thyroidectomy with or without lymph node (LN) dissection according to presurgical findings on U/S and with or without RAI ablation); the number of lesions (cervical LNs and/or malignant soft tissue) ≤ 2, the size of lesions ≤ 8 mm (central compartment) and ≤10 mm (lateral compartments) in the smallest diameter; patient’s unwillingness to undergo AS or revision surgery; written informed consent to undergo RFA with the annotations that the optimal treatment for cervical recurrent/persistent disease is surgery and AS is a reasonable choice in cases of smaller lesions; and the existence of other malignant cervical lesions not detected by high-resolution U/S cannot be excluded. The exclusion criterion was the presence of distant metastases as confirmed by anatomic imaging (post-RAI whole-body scan (WBS), CT scan, MRI, Fludeoxyglucose-18 (FDG), positron emission tomography (PET)—CT scan). Persistent or recurrent disease was defined as SIR < 12 or ≥12 months, respectively, after initial surgical treatment [23]. The following information was extracted from patients’ medical records: demographics, clinical, histopathological and biochemical data, therapeutic interventions and complications of treatment. Patients were followed with a biochemical profile (Thyroid-stimulating hormone (TSH), Free Thyroxine (FT4), Thyroglobulin (Tg), Antithyroglobulin antibodies (Anti-Tgs)) and high-resolution U/S at regular time intervals of one month, three months and then every three months, after RFA was performed.

### 2.2. Biochemistry

Measurements of thyroid biochemical parameters (TSH, FT4, Tg, Anti-Tg) were made on blood samples collected between 8:00 and 9:00 AM prior to RFA and during follow-up by a Beckman Coulter RIA/IRMA (Radioimmunoassay/Immunoradiometric Assay) KIT, according to standard laboratory protocols. The same KIT was used to evaluate Tg levels of washout fluid.

### 2.3. RFA Procedure

The RFA procedure was performed at the Day Clinic of Saint Savvas Anticancer Oncological Hospital of Athens by the same experienced interventional radiologist. Patients were placed in a supine position with their neck fully extended. Targeted lesions and anatomic structures were visualized by U/S in real time using a GE LOGIQ P9 U/S system with a linear probe operating at a frequency of 6 to 14 MHz. The volume of targeted lesions was calculated using the following equation:V = πabc/6V is the volume, a is the largest diameter and b and c are the two other perpendicular diameters.

Local anesthesia with lidocaine (2% cream) was applied for mild local analgesia after routine disinfection. RFA was performed with the moving shot technique after hydrodissection using cold (0–4 °C) dextrose 5% in water (D5W), which was injected between the targeted lesion and vital structures to create a safe zone, thus preventing thermal injury. A 19-gauge internally cooled electrode (STARMed, Seoul, South Korea) that was 7 cm in length with a 0.5 cm active tip was used, powered by a VIVA RF generator (STARMed). Voice evaluation was assessed by asking patients to verbalize during the procedure; the patient’s vital signs were continuously observed, accordingly. The energy applied per unit volume (E/V) was calculated as follows:E/V (J/mL) = [Power (watts) × Active ablation time (seconds)]/Volume of LN(s) (mL).

Ablation was completed when the targeted lesion became hyperechoic. Consequently, every patient was hospitalized in the short stay unit (SSU) for at least 2 h post-ablation to evaluate for possible short-term minor or major complications.

### 2.4. Statistical Analysis

Descriptive data are expressed as the mean ± standard deviation (SD) for normally distributed variables; otherwise, the median value and interquartile range (IQR) are shown. To assess the normal distribution of the data, we used the Shapiro–Wilk test. The Wilcoxon signed rank test was used to compare tumor volume and serum Tg concentrations before RFA and at the last follow-up visit. The level of significance was defined as *p* < 0.05. All analyses were conducted using SPSS (version 24, IBM Corp., SPSS, Chicago, IL, USA).

## 3. Results

Clinical characteristics and histopathological data of the eight enrolled patients (six women/two men) at baseline are illustrated in Table 1. The mean age was 39.3 ± 16.4 years (range: 17–69). The median time interval between the last surgery for every patient and the documentation of persistence/recurrence by high-resolution U/S was 22.5 (IQR: 39) months (range: 3–66). All patients prior to RFA had undergone one surgery (total thyroidectomy with or without LN dissection according to the presurgical findings as were documented in high-resolution U/S) except for one (patient No2), who had undergone three surgeries due to consecutive locoregional recurrences in the thyroid bed concomitantly with biochemical recurrence, albeit the initial excellent response to treatment (i.e., thyroidectomy and RAI ablation with negative imaging in WBS and undetectable Tg levels). It is worth mentioning that the first recurrence of patient No2 was documented 34 months after the first surgery was performed, while the second recurrence was documented 13 months after the first recurrence. The mean number of RAI ablation treatments (RAI-A-T) was 1.5 ± 0.93 (range, 0–3); no RAI-A-T in one, one RAI-A-T in three, two RAI-A-T in three, and three RAI-A-T in one patient) with a mean RAI activity of 160 ± 107.97 mCi (range 0–300). Two of the patients were diagnosed with aggressive histological variants; patient No2: oncocytic (OCA) widely invasive, and patient No3: follicular with trabecular/insular/solid patterns. LN infiltration was already present in five patients at the time of diagnosis.

In total, 14 malignant lesions were ablated by RF. The mean number of ablated lesions per patient was 1.75 ± 0.7 (range: 1–3). Four patients were treated for two lesions and three patients were treated for one lesion. One patient (patient No2) was initially treated for two lesions and three months after the first RFA treatment; a new lesion was detected and was treated successfully by a second RFA. Six lesions located in the central (level VI) and eight in the lateral compartments (levels III and IV). Mean largest and smallest lesion diameters were 10.67 ± 1.99 mm (range: 8–13.9) and 5.69 ± 1.68 mm (range: 3.2–9.2), respectively. The median lesion volume was 0.24 mL (IQR: 0.24, range: 0.09–0.9). The power used for ablation ranged from 5 to 20 W (median: 10, IQR: 8.75), and the ablation time ranged from 98 to 948 s (median: 300, IQR: 254). The energy delivered per mL of pretreatment lesion ranged from 1633.33 to 13542.9 J/mL (mean: 7594.9, SD: 4867.2), and the total energy delivered ranged from 670 J to 40160 J (median: 18410, IQR: 28870). The mean follow-up period from the time that the first RFA was performed to the last visit was 13.25 ± 7.5 months (range: 4–24).

We recorded the lesions’ volume reduction after RFA was performed. The median volume of the lesions reduced significantly from 0.24 mL (IQR: 0.24, range: 0.09–0.9) to 0.02 (IQR: 0.01, range: 0–0.03), (*p* = 0.001) with a median volume reduction ratio (VRR) of 94.5% (IQR: 3.25, range: 78–100) (Table 2 and Table 3). The median VRR increased from 67% in 1 month to 93% in the 12-month follow-up (Figure 1). The volume reduction ratio was calculated as follows:VRR = [baseline volume (mL) − final volume (mL)]/baseline volume (mL) × 100.

Indicatively, the targeted lesion of patient No4 (i.e., the patient with the longer f-up) is depicted in Figure 2a–c, before RFA, at one month after RFA, and at the final f-up 24 months after RFA, respectively. Out of 14 lesions that were treated, 2 (14.3%) were no longer visible in the U/S.

All patients underwent a single session of RFA except patient Νο2, who received an additional RFA for a new recurrence. Meanwhile, in the same patient, re-recurrence of the two lesions that were treated with RFA was documented 9 months after the first RFA. Both lesions were re-treated successfully with a second RFA (Figure 3).

TSH levels in the whole cohort ranged between 0.1 and 0.5 mIU/L throughout the follow-up period, while median Tg levels at baseline were 1.05 ng/mL (IQR: 6.64, range 0.2–9.97) and reduced significantly to 0.2 ng/mL (IQR: 0.34, range: 0–0.5), (*p* = 0.028) at the end of follow-up (Table 2 and Table 3). Anti-Tg antibodies were negative in all but two patients (No3 and No8); in these two patients, anti-Tgs reduced, from 88 times the upper reference limit (URL) to 40 times the URL and from 4.9 times the URL to 3.5 times the URL, respectively. Nevertheless, in patient No3, a rise in anti-Tg levels was documented at the last follow-up (21 months after first RFA), which was subsequently followed by the detection of two new locoregional recurrences (0.24 mL and 0.07 mL) plus two distant metastases measuring up to 9 mm at the left lower lung lobe.

During the treatment, two patients developed Horner syndrome with unilateral myosis, ptosis and anhidrosis as a complication of RFA. In Figure 4a,b, the initial presentation of the syndrome in patient No1 and the significant improvement after a period of 14 days is depicted; the apraclonidine test causing reversal anisocoria was used to establish the diagnosis (Figure 5) [43]. Glucocorticoid administration (0.5 mg/kg/d) was initiated for 7 days followed by tapering the dose over 3 days. Complete resolution of the signs and symptoms was documented within 6 months. There were no other significant complications or voice changes except for tolerable pain or a burning sensation during the RFA treatment. In general, all the patients tolerated the RFA procedure well.

## 4. Discussion

WDTC is characterized by an excellent prognosis and no need for long-term follow-up or additional diagnostic and therapeutic interventions in the majority of cases, are needed. Nevertheless, SIR (i.e., locoregional recurrence or disease persistence) can be encountered in a subgroup of patients, especially when they are diagnosed with an aggressive histology and classified as ATA intermediate and high risk [19,20,21,22]. Surgery is the standard of care, in cases where the malignant lesion is resectable. RAI ablation may be considered as adjuvant treatment in RAI-avid lesions usually after revision surgery has been performed, albeit it has not shown superiority regarding prognosis and recurrence-free survival, with the ATA-risk classification and histology as the most important prognostic factors of response to treatment [44,45]. Similarly, the reoperation for SIR has shown inconclusive results with a range of efficacy between 40% and 100%, mainly because of not well-defined criteria for patient selection and determination of a successful surgery; moreover, complications from reoperation should be considered in the decision-making process [46]. On the other hand, in cases of smaller or/and iodide refractory local metastases, patient reluctance or inability to undergo revision surgery MIT, like RFA, may be considered [19,24]. In the present study, we have aimed to evaluate the safety and efficacy of RFA in a cohort of mainly ATA intermediate risk Caucasian patients who experienced locoregional disease recurrence/persistence.

In our cohort, RFA seems to be an efficient MIT achieving a high level of structural response with a median VRR of 94.5%. Interestingly, two lesions of two different patients were no longer visible in the U/S. Accordingly, biochemical remission was documented with a significant decrease in serum Tg and Anti-Tg levels in all but one patient (patient No3). Our results are supported from a large number of studies in the literature where RFA has been shown to be an effective therapeutic modality in recurrent DTC with a VRR ranging from 53% to 100% [33,34,36,38,47]. Systematic reviews and meta-analyses confirm the safety and effectiveness of the method in reducing the lesion volume and serum Tg levels as well, with some studies reporting the complete eradication of the targeted lesions in a percentage of up to 92% after 24 months of follow-up [32,39,48].

Recent studies with extended follow-up support the aforementioned findings, as well. In a Chinese cohort of 32 patients and 58 locoregional recurrent PTC lesions, all but one lesion had been completely eradicated at the end of the follow-up period, which was not earlier than 60 months post-ablation [37]. In a large retrospective analysis including 119 patients with 172 locoregional recurrent DTC lesions, the VRR after RFA was 81.2%, with 72.1% of the lesions completely disappearing after a mean follow-up period of 47.9 months. Lesions characterized by invasion into the airways demonstrated the most unfavorable prognostic outcomes. Nevertheless, even in such cases, RFA should be considered as 50% of the lesions invading the trachea were completely eradicated [49]. It should be noted that RFA may be considered in small and not large malignant lesions, according to the international guidelines and recommendations [19,24]. In a Taiwanese cohort of 23 patients with 52 locoregional recurrent DTC lesions, 29 out of 52 lesions (55.8%) had completely disappeared at a follow-up of a 6-month period with a mean VRR of 86.6%. Nevertheless, lesions with a maximum diameter exceeding 3.2 cm prior to the RFA demonstrated the least favorable therapeutic outcomes [47].

One of our patients (patient No2) experienced a re-recurrence, which was successfully treated with a revision of the RFA. Re-recurrences after RFA had also been described in the literature during long-term follow-ups with a revision RFA achieving the adequate control of tumor growth [36,37]. Especially OCA, previously called “Hürthle cell”, is characterized by locoregional metastases and RAI refractoriness as it is our case. Notably, this patient had experienced local recurrencies twice, before the initial RFA, despite the preceded successful surgeries and adjuvant RAI-A-T with negative post-therapy WBS. Tailoring of the monitoring and treatment is of great importance towards the best therapeutic decision in this aggressive type of TC [50].

In one of our patients (patient No3), two new recurrences were documented at the end of the follow-up period, while distant metastases were revealed in 18F-FDG PET-CT scan. This patient was diagnosed with FTC presenting trabecular/insular/solid patterns, histopathological characteristics towards the dedifferentiation process and aggressive biological behavior [9,15]. This patient had been treated twice with RAI before RFA was performed, and no uptake in WBS was documented, even in the thyroid bed, where recurrences were documented by high-resolution U/S. Due to older age (69 years old) and comorbidities, the patient preferred RFA instead of surgery. In general, locoregional SIR in ATA intermediate and high-risk patients is an independent prognostic factor for progressive metastatic disease [19,51]. In such cases, it is of great importance to balance the risk/benefit ratio of any therapeutic modality in order to “*first do no harm*”. Such patients are candidates for TKIs (Tyrosine Kinase Inhibitors) treatment but no sooner than they fulfill the RECIST (Response Evaluation Criteria in Solid Tumors) criteria for disease progression and/or locoregional disease poses a threat for vital structures (i.e., trachea). Any kind of locoregional therapies, including RFA, comes first to the therapeutic arsenal before the initiation of systemic treatment [19].

Finally, in two of our patients, Horner syndrome was diagnosed as a short-term complication of RFA. Horner syndrome is a very rare side effect of RFA described only in a very few case reports of the literature [52,53]. It is resolved in a short period of time without leaving any permanent signs or symptoms. It can be rarely present after thyroidectomy or revision surgery for locoregional disease, as well [54,55].

Our study has strengths and some limitations. We have included Caucasian patients with both persistent and recurrent disease, followed up for a reasonable period of time; most studies in the literature come from Asia; thus, it is of value to collect data from a European country and, far to our knowledge, this is the first study to be published regarding data from Greece. Moreover, we included patients with re-recurrence and new recurrence as well, studying the efficacy and safety of RFA in these cases. Finally, the inclusion of patients with aggressive histology is of great importance as data regarding these patients and appropriate therapeutic management are still gathered. Regarding the limitations, the most obvious is that of the small sample size. This study was further limited by its retrospective design.

## 5. Conclusions

Despite the small sample size, we showed that RFA is an effective and safe therapeutic modality to be considered in selective DTC cases with locoregional SIR (persistent/recurrent disease), achieving a VRR of 94.5% and subsequent biochemical remission. Smaller lesions and DTC cases characterized by the increased risk of local re-recurrence (i.e., aggressive RAI refractory histological variants) may be considered as candidates for this type of MIT, providing that the patient has meticulously been informed about the risk/benefit ratio of all the available therapeutic choices. Additional prospective studies with longer follow-up periods and larger sample size should be performed towards a tailored therapeutic approach, especially in more aggressive histological variants of TC.

## Figures and Tables

**Figure 1 biomedicines-13-00255-f001:**
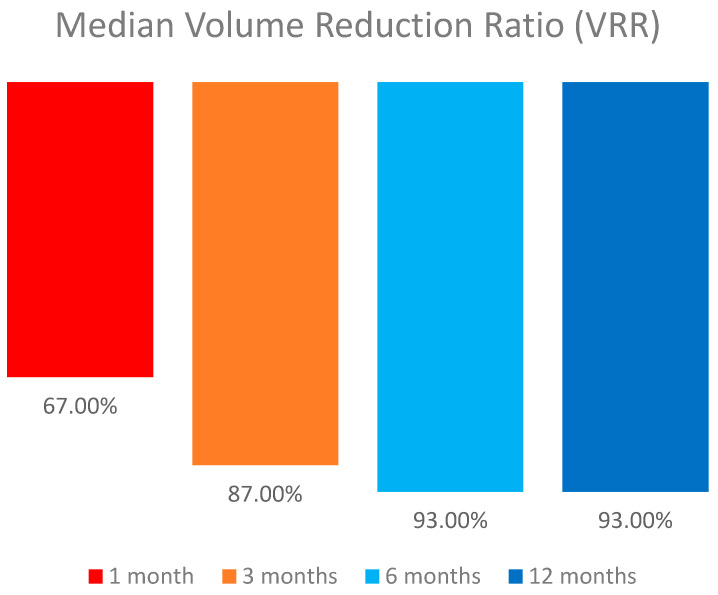
Graph shows median volume reduction ratio (VRR%) at each follow-up visit.

**Figure 2 biomedicines-13-00255-f002:**
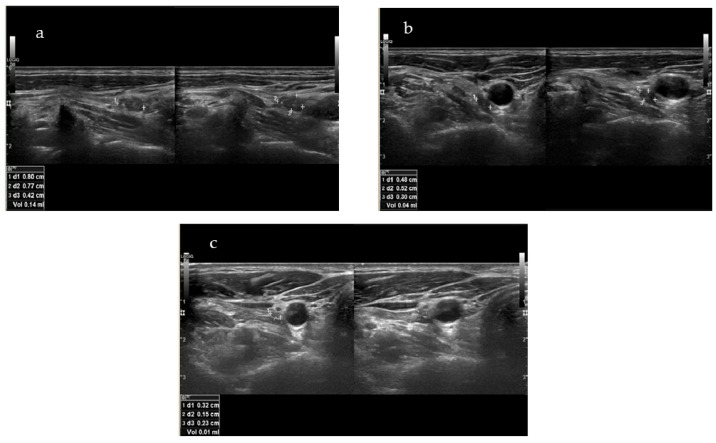
(**a**–**c**) Targeted lesion of patient No4 before RFA (**a**), at one month after RFA (**b**) and at final f-up 24 months after RFA (**c**).

**Figure 3 biomedicines-13-00255-f003:**
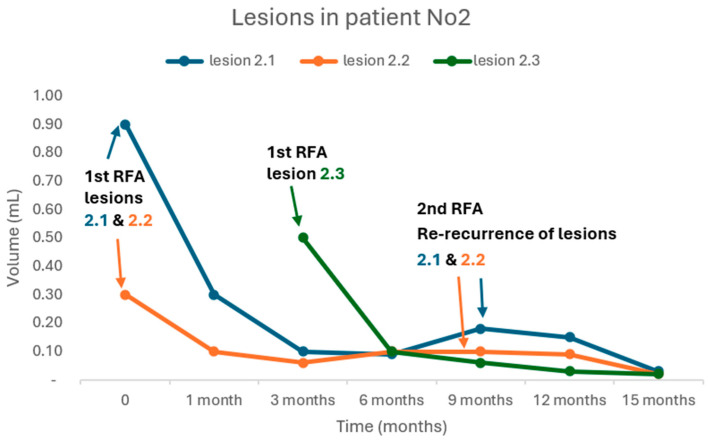
Targeted lesions in patient No2: RFA: radiofrequency ablation. Lesion 2.3 occurred 3 months after the 1st RFA was performed to lesions 2.1 and 2.2. Lesions 2.1 and 2.2 recurred 9 months after they were RF-ablated. A 2nd RFA was performed.

**Figure 4 biomedicines-13-00255-f004:**
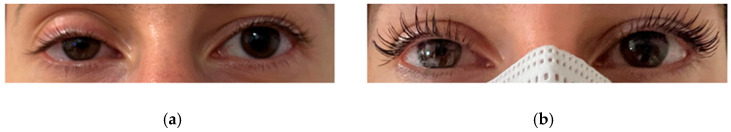
(**a**,**b**) Initial presentation of Horner syndrome in patient No1 and the significant improvement after a period of 14 days with glucocorticoid administration (0.5 mg/kg/d) for 7 days followed by tapering the dose over 3 days.

**Figure 5 biomedicines-13-00255-f005:**
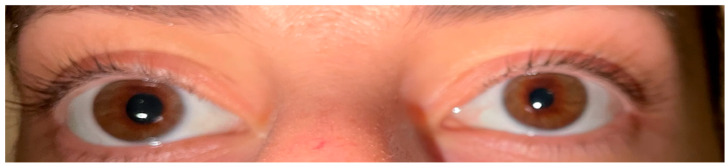
Horner syndrome documented with the apraclonidine test causing reversal anisocoria.

**Table 1 biomedicines-13-00255-t001:** Baseline characteristics and histopathological data of the patients.

Patients (Age/Sex)	ATA Risk Classification	T N M	Histology	No of Surg.	No of RAI- A-T (mCi)	Disease	Lesion/ Loc.	Largest Diam. (mm)	Time (Months)
Pt No1 (27/F)	low	T2 N0 M0	PTC (classical variant)	1	1 (150)	R	LN/III LN/IV	12.1 11.5	51
Pt No2 (45/M)	intermediate	T3b N0 M0	OCA widely invasive	3	3 (300)	R	TIS/VI TIS/VI	13.9 9.6	34
							TIS/VI	12.8	66
Pt No3 (69/F)	intermediate	T3b N1b M0	FTC with trabecular/ insular/ solid patterns	1	2 (220)	R	LN/IV TIS/VI	10.6 8.8	39
							LN/IV LN/IV	7.9 * 6.6 *	65
Pt No4 (17/M)	intermediate	T1a(m) N1 M0	PTC (classical variant)	1	1 (90)	P	LN/IV	8	3
Pt No5 (45/F)	intermediate	T3b(m) N0 M0	PTC (classical variant)	1	0 (0)	P	LN/IV LN/IV	11.5 9.7	3
Pt No6 (50/F)	intermediate	T3b N1a M0	PTC (classical variant)	1	1 (70)	P	LN/VI LN/IV	8.2 13.2	3
Pt No7 (34/F)	intermediate	T3b(m) N1a M0	PTC (tall-cell variant)	1	2 (300)	P	LN/VI (delphian)	8	4
Pt No8 (27/F)	intermediate	T2 N1a M0	PTC (classical variant)	1	2 (150)	P	LN/III	11.5	11

Age is given in years at time of thyroid cancer (TC) diagnosis. F: female, M: male. ATA: American Thyroid Association. PTC: papillary thyroid carcinoma. OCA: oncocytic carcinoma. FTC: follicular thyroid carcinoma. Surg.: surgeries, refer to the total number of surgeries the patient has undergone prior to RFA. R: recurrent disease (structural incomplete response ≥ 12 months after surgery). P: persistent disease (structural incomplete response < 12 months after surgery). Lesion: type of malignant tissue (LN for lymph node or TIS for soft tissue). Loc.: location, refers to where the neck compartment lesion was located. Largest diam. (diameter) refers to the largest initial diameter of the lesion. Time refers to time from 1st surgery to 1st recurrence. (*): The two lesions of patient No3 marked with (*) have not been treated yet; two additional lung metastases were documented in this patient (see text in the results section).

**Table 2 biomedicines-13-00255-t002:** Lesion volume and biochemical characteristics prior to RFA and at the end of follow-up.

Patients	Pre-RFA Volume (mL)	Pre-RFA Tg (ng/mL) or Anti-Tg	Post-RFA Volume (mL)	Post-RFA Tg (ng/mL) or Anti-Tg	Time (Months)	VRR (%)
Pt No1	lesion 1.1: 0.4 lesion 1.2: 0.2	0.2	0 0.01	<0.1	12	100 95
Pt No2	lesion 2.1: 0.9 lesion 2.2: 0.3	9.97	0.03 0.02	0.2 0.2	15	96 93
	lesion 2.3: 0.5 ^†^	6.2	0.03	0.2	12	94
Pt No3	lesion 3.1: 0.5 lesion 3.2: 0.2	88 *	0 0.01	40 *	21	100 95
Pt No4	0.14	5.8	0.01	0.5	24	93
Pt No5	lesion 5.1: 0.4 lesion 5.1: 0.2	1.2	0.02 0.01	0.28	6	95 95
Pt No6	lesion 6.1: 0.12 lesion 6.2: 0.24	0.21	0.01 0.02	<0.1	18	92 92
Pt No7	0.09	0.9	0.02	0.2	6	78
Pt No8	0.23	4.9 *	0.03	3.5 *	4	87

RFA: radiofrequency ablation, time refers to follow-up time from the first RFA to last visit. *: anti-Tg values refer to x times the upper reference limit (URL). ^†^ Lesion 2.3 occurred 3 months after RFA was performed in lesions 2.1 and 2.2.

**Table 3 biomedicines-13-00255-t003:** Median changes in lesion volume and patients’ Tg from the first RFA to the last f-up visit.

	Pre-RFA Median (IQR)	Post-RFA Median (IQR)	*p*
Volume (mL)	0.24	0.02	0.001
Thyroglobulin (ng/mL)	1.05	0.2	0.028

## Data Availability

Data are contained within this article.

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
