# Peer review of "Radiofrequency Ablation for Locoregional Structural Incomplete Response in Differentiated Thyroid Cancer: Initial Experience in Greece"

_biomedicines, 2025, doi:10.3390/biomedicines13020255_

Round 1
Reviewer 1 Report
Comments and Suggestions for Authors
This retrospective study investigates the effectiveness and safety of RFA in patients with DTC who have SIR. The results indicate that RFA has clinical value for treating SIR in DTC patients; however, there are still some issues that need to be addressed in this manuscript:
1.The research sample size is too small
2.No comparison of images before and after ablation, as well as images of the patient after recovery of Hornor syndrome, were provided.
3.Please pay attention to many writing details, such as some units not being written(Page 3, line 125), and some abbreviations appearing in capital letters for the first time without their full names(Page 1, line 30)
4.Out of 6 patients, 2 of them developed Hornor syndrome. How does this compare to the incidence of complications in previous surgical patients?
5.The research sample size is small, and some patients have a follow-up time of less than 12 months. Their loss to follow-up will have a significant impact on follow-up data such as VRR.
Author Response
Comment 1: The research sample size is too small
Response 1: Thank you for pointing this out. We have mentioned the small sample size as one of the major limitations of our study. Nevertheless, we should take into consideration, as it has already been mentioned in the main text, that RFA is not widely used in the western countries. Thus, it is not easy to have a large cohort. Moreover, regarding Greece, is the first ever study trying to elucidate the safety and efficacy of RFA in thyroid cancer patients with recurrent/persistent disease and our center is the only one in the whole country performing this procedure in recurrent/persistent disease in thyroid cancer patients.
Comment 2: No comparison of images before and after ablation, as well as images of the patient after recovery of Hornor syndrome, were provided.
Response 2: Thank you for your comment. We have now added additional information in the text, page 6, lines 239-241 and page 9, lines 290-292. Moreover pictures of patient No4 (i.e the patient with the longer follow-up of 24 months) are depicted in page 8, before RFA, at one month after RFA and at final f-up 24 months after RFA. Additionally, pictures of one of two patients diagnosed with Horner syndrome are depicted in page 9, at time of diagnosis and 14 days after with significant improvement. Unfortunately we do not have the picture of the young lady after one month where the syndrome has completely been resolved.
Comment 3: Please pay attention to many writing details, such as some units not being written(Page 3, line 125), and some abbreviations appearing in capital letters for the first time without their full names(Page 1, line 30)
Response 3: Thank you for your comment. We have corrected the mistakes accordingly (page 1 lines 30-31). Regarding what you mention in page 3 line 125 , the number is reffering to the number of lesions, so no units are required.
Comment 4: Out of 6 patients, 2 of them developed Hornor syndrome. How does this compare to the incidence of complications in previous surgical patients?
Response 4: Thank you again for your useful comment. Out of 8 patients 2 of them developed Horner syndrome. There is no head to head study comparing the complications in patients undergoing revision surgery and on the other hand selecting RFA. It would be of value a study like that in large cohort of patients perfectly matched regarding age, histopathology, similarity of anatomical structures etc. Moreover the aim of our study was to present original data regarding safety and efficacy of RFA, even in patients with aggressive histology and patients who have already underwent surgical treatment.
Comment 5: The research sample size is small, and some patients have a follow-up time of less than 12 months. Their loss to follow-up will have a significant impact on follow-up data such as VRR.
Response 5: Thank you for pointing this out. As we have already mentioned in the manuscript and in our response 1, the sample size is small but we should take into consideration, as it has already been mentioned in the main text, that RFA is not widely used in the western countries. No patient has been lost to follow-up. As it is a retrospective study we aimed to publish our data till the time of publication. Some patients have been followed-up for a period of less than 12 months because their first referral to our department was within 2024.
Reviewer 2 Report
Comments and Suggestions for Authors
The manuscript is well written, it is easily understood, the use of English language is appropriate.
There are a few misspellings that need to be corrected, but nothing major.
I only have one suggestion for the improvement of the text, line 50, I do not understand the numbers, on which population do they refer? World, USA? Please correct this.
Author Response
Comment 1: The manuscript is well written, it is easily understood, the use of English language is appropriate. There are a few misspellings that need to be corrected, but nothing major. I only have one suggestion for the improvement of the text, line 50, I do not understand the numbers, on which population do they refer? World, USA? Please correct this.
Response 1: Thank you for your kind words regarding the initial experience of Radiofrequency Ablation for Locoregional Structural Incomplete Response in Differentiated Thyroid Cancer, in Greece. We have checked again the manuscript for misspellings and we corrected them. Regarding your question an additional phrase has now been added in the manuscript, page 2, line 49.
Round 2
Reviewer 1 Report
Comments and Suggestions for Authors
I appreciate the effort you've put into addressing the comments. The overall structure of the paper is solid, and the revisions have notably improved its clarity and quality. For future research, I suggest expanding the sample size to strengthen the robustness of the data and enhance the generalizability of your findings.